# Statistical analysis of the impact of environmental temperature on the exponential growth rate of cases infected by COVID-19

**George Livadiotis** *

Southwest Research Institute, San Antonio, TX, United States of America

* glivadiotis@swri.edu

**Data Availability Statement:** All relevant data are within the manuscript.

**Funding:** The author(s) received no specific funding for this work.

## Abstract

We perform a statistical analysis for understanding the effect of the environmental temperature on the exponential growth rate of the cases infected by COVID-19 for US and Italian regions. In particular, we analyze the datasets of regional infected cases, derive the growth rates for regions characterized by a readable exponential growth phase in their evolution spread curve and plot them against the environmental temperatures averaged within the same regions, derive the relationship between temperature and growth rate, and evaluate its statistical confidence. The results clearly support the first reported statistically significant relationship of negative correlation between the average environmental temperature and exponential growth rates of the infected cases. The critical temperature, which eliminates the exponential growth, and thus the COVID-19 spread in US regions, is estimated to be $T_C = 86.1 \pm 4.3$ F$^O$.

## 1. Introduction

The daily number of new cases infected by COVID-19 is currently exponentially growing for most countries affected by the virus. However, this exponential growth rate varies significantly for different regions over the globe. It is urgent and timely to understand the reasons behind this regional variation of the exponential growth rates. Little information is known about this matter, while there are indications that the environmental temperature may be a factor; for instance, northern and colder US and Italian regions experienced much more incidents than others.

Typically, the evolution curve of the spread of the coronavirus initiates with a pre-exponential phase, which is characterized by a mild logarithmic growth, followed by the outbreak, that is, the phase of the exponential growth. Social-distancing measures against the spread may affect the evolution curve in a way that the exponential growth slows down (decelerated phase) and starts to decline (decline or decay phase [1]), depending though on the effectiveness and applicability of these measures. However, after the decline of the spread at some place, new infected cases may outbreak in other places, marked with insignificant number of cases until

**Competing interests:** The authors have declared that no competing interests exist.

that moment. Then, a newly growth phase may appear. For example, Fig 1 (left) shows the evolution curve of the spread for the infected cases in mainland China; clearly, we observe the whole growth–decay cycle, as well as, a new re-growth phase.

Super-strict measures, such as complete shut down and quarantines, can successfully lead to the deceleration of the exponential growth of infected cases [2]. Unfortunately, they cannot be successfully applied and followed within vast regions, and especially, for a long and indefinite period of time. Inevitably, measures may be loosened during the decay phase – if not earlier, leading to the birth of an equally disastrous re-growth phase.

The exponential growth is the most effective phase for the evolution curve of infected cases; and the most important question regarding this evolution is still open [3]: *What can influence the exponential growth rate, and thus, "flatten the curve"*? Measures, strict or not, may affect the evolution of new infected cases, by shifting the spread curve from the exponential to the decelerated growth. It should be noted though that measures do not affect the exponential growth rate itself, but only the period of time that this exponential phase applies. Then, what factors do affect the exponential growth rate?

The age distribution in the place where the outbreak occurs is unlikely to be a factor; indeed, the number of new cases is known to be positively correlated with age, however, the exponential growth rate (China: 0.169; US: 0.121; Italy: 0.090 –decreasing rate) appears to be negatively correlated to the age median of these countries (China: 37.4; US: 38.1; Italy: 45.5 – increasing age); hence, the age is likely irrelevant to the rate variations.

In addition, culture in social activities may be a factor; for example, this might be contributing in the observed differences among the exponential rates in the cases of China, Italy, and US (Fig 1). However, what is causing the major variation of exponential rates among different regions of the same culture? It is apparent that culture does not constitute the main factor influencing the exponential rate.

Fig 2 shows the regional variation of infected cases (left) during the exponential growth phase and the average winter temperature (right) in Italy. The possible negative correlation, observed between regional number of infected cases and winter temperature in Italy, is an indication of the influence of temperature on the exponential growth, but it certainly does not constitute a necessary condition. The reason is that the map plots the total number of the infected cases $N_t$, which is not dependent only on the exponential growth rate $\lambda$, but also on the initial number of cases $N_0$.

It is generally accepted that the initial infected cases in Italy were travelled directly from China; since some destinations are more favorable than others, then, the initial number of cases $N_0$, as well as the current number of cases $N_t$ (which is proportional to $N_0$), should be subject of regional variation. Therefore, there is a non-negligible possibility, the observed regional variation of the number of infected cases $N_t$ to be caused by the regional distribution of the initial cases $N_0$. In such a case, main airport cities would have incredibly high number of infected cases outplaying a possible negative correlation of daily infected cases with regional average temperature $T$; the latter may be one of the reasons of the high numbers of cases observed in New York City and Rome.

On the other hand, in their letter to the White House, members of a National Academy of Sciences committee said that "*There is some evidence to suggest that [coronavirus] may transmit less efficiently in environments with higher ambient temperature and humidity; however, given the lack of host immunity globally, this reduction in transmission efficiency may not lead to a significant reduction in disease spread without the concomitant adoption of major public health interventions*" [4].

Nevertheless, it has to be stressed out that there were no statistical analyses focused on the exponential growth rates of the infected cases in regions with different temperatures. For

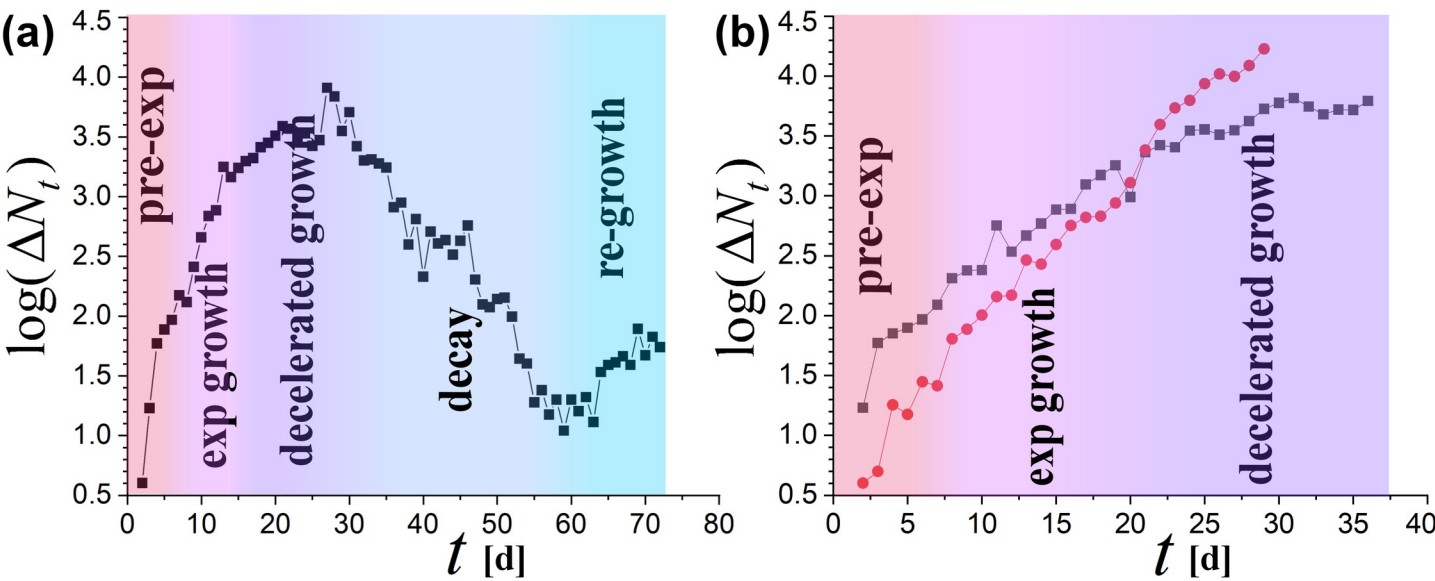

**Fig 1. Evolution of new cases infected by COVID-19 (on linear-log scale) for mainland China (left) and US & Italy (right); phases (color-coded): Pre-exponential (pre-exp), exponential (exp) growth, decelerated growth, decay (or decline), and possibly, a re-growth**. Day $t = 1$ corresponds to 1/15/2020 for China, 2/20/2020 for Italy, 2/27/2020 for US. Evolution in China cases follows the whole growth-decay cycle, and a new re-growth phase. Italian cases are characterized by a milder exponential rate, entered the phase of decelerated growth on March 12. US suffers with a larger exponential rate, and it is not clear whether has entered the decelerated growth phase. The exponential growth rate for China rose as high as $\lambda = 0.169\pm0.017$, while for Italy and US the rates were $\lambda = 0.090\pm0.004$ and $0.121\pm0.003$, respectively (with correlation coefficient > 0.99).

instance, several authors (e.g., [5, 6]) found insignificant correlations between temperatures and confirmed cases. However, their analysis was performed on the number of the infected cases $N_t$, which is subject to the randomness of the initial cases $N_0$ as explained above, and not

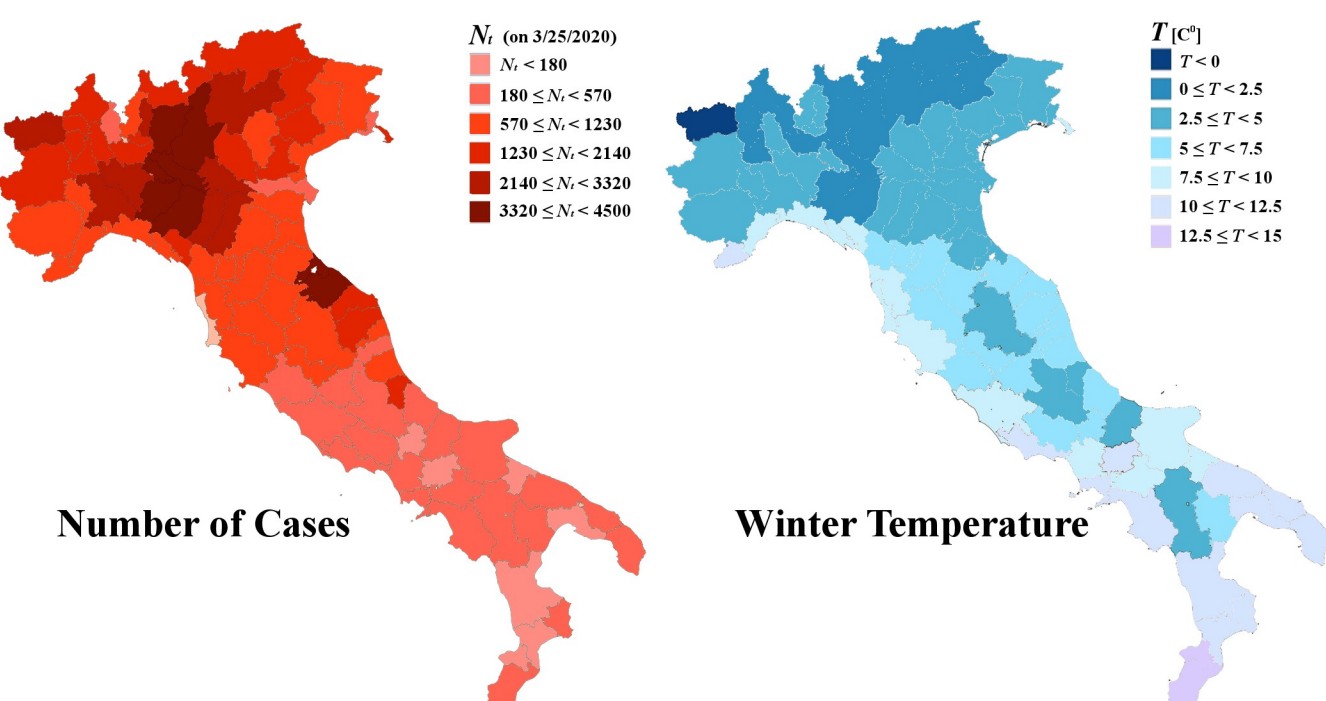

**Fig 2.** Regional distribution of infected cases $N_t$ by 3/25/2020 (left), and average winter temperature $T$ in mainland Italy (right).

on the exponential growth rate λ, which is clearly dependent on physical characteristics of the coronavirus, binding protein, and environment.

Analysis of regional cases can show whether the speculated negative correlation between temperature and number of infected cases is true, meaning a negative correlation between temperature and exponential growth rate. If the environmental temperature plays indeed a substantial role on the virus spread, then, this can provide promising results, such as, the estimation of the critical temperature that may eliminate the number of daily new cases in heavily infected regions.

The purpose of this paper is to improve our understanding of the effect of environmental temperature on the spread of COVID-19 and its exponential growth rate. In particular, we calculate the exponential growth rates of infected cases for US and Italian regions, derive the relationship of these rates with the environmental temperature, evaluate its statistical confidence, and determine the critical temperature that eliminates this rate.

## 2. Theory

### 2.1 Modeling behind "flattening the curve"

A standard model for describing the evolution of the infected cases by viruses can be constructed as follows

$$\frac{dx}{dt} = E(x) \cdot I(x), \text{ with } E(x) = \lambda \cdot x, \ I(x) = 1 - x^b, \tag{1}$$

where $x(t) \equiv N(t)/N_{max}$, $x_0 \equiv N_0/N_{max}$; $N(t)$ is the number of total infected cases evolved from the initial $N_0 \equiv N(0)$ cases, $N_{max}$ is the maximum possible number of infected cass; λ is the exponential growth rate, and becomes clear for $x(t) << 1$ where $I$ is negligible, leading to:

$$x(t) = x_0 \cdot \exp(\lambda t) \text{ or } N(t) = N_0 \cdot \exp(\lambda t), \text{ for } x << 1 \text{ where } I(x) \cong 1. \tag{2A}$$

Typically, data of infected cases are daily provided and updated. Thus we set the readout of $N(t)$ on a daily basis, such as: $N_t \equiv N(t/[d])$. Hence, we may write

$$x_t = x_0 \cdot \exp(\lambda t) \text{ or } N_t = N_0 \cdot \exp(\lambda t), \tag{2B}$$

with where $x_t \equiv N_t/N_{max}$ with $t$ indicating the time on a daily basis, ($t$ = 1d, 2d, . . .).

The function of negative feedback $I$ models the factors that flattens the curve, such as, the measures taken against spreading. While these factors are not affecting the exponential growth rate λ, they become more effective as the number of cases increases, getting closer to $N_{max}$; exponent $b$ controls the effectiveness of these factors; strict {loose} measures correspond to smaller {larger} values of $b$.

Fig 3 shows the evolution curve of the number of new ($\Delta N_t = N_{t+1} - N_t$) and total infected cases ($N_t$), as well as, how this curve flattens for stricter measures (i.e., smaller values of $b$).

As observed in Fig 3(A), stricter measures, nicely modeled by decreasing $b$, do not affect the exponential rate λ but they successfully flatten the curve. However, the same can be achieved by downgrading the exponential growth rate, as shown in Fig 3(B).

It is apparent, then, how much useful would it be to know the factors that can flatten the curve by decreasing directly the exponential growth rate, instead of applying stricter measures. Applied measures could be loosen and shorter!

Model (1) originates from the logistic map family (e.g., [7], and references therein; [8]); other complicate versions, such as, the Susceptible–Infectious–Recovered models (e.g., [9]) may be expressed by multi-dimensional differential or difference equations (e.g., [10], and references therein; [11]), but still, the curve flattening is governed by the same features. The two composites, the exponential growth $E$ and the negative feedback $I$, are just the main and

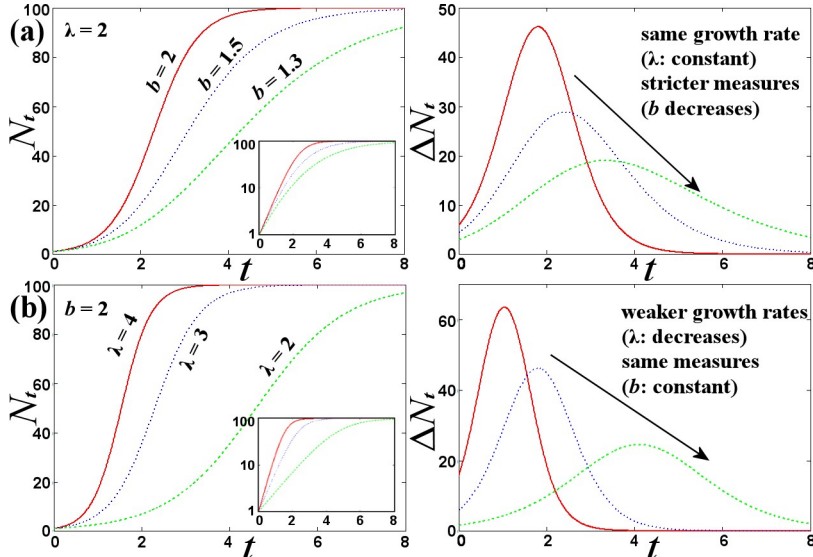

**Fig 3.** Plots of total $N_t$ and new $\Delta N_t$ infected cases according to model (1), showing the flattening of the spread curve with the interplay of (a) stricter measures (decreasing $b$), or (b) weaker rates (decreasing $\lambda$). (Input panels show graphs on semi-log scale.).

necessary conditions for reproducing the growth-decay phases of the spread curve. Their interplay shows how the spread curve can be flattened as a result of stricter measures, independently of the existent exponential rate.

## 2.2 Main factors influencing the exponential growth rate

What are the main factors that can affect the exponential growth rate $\lambda$ of COVID-19 spread?

The rate $\lambda$ is expected to have positive correlation with the reproduction number $R_0$ (e.g., proportional to its logarithm), and negative correlation with the incubation period $\tau$ (e.g., inverse proportional) [12]. The number $R_0$ is a measure of how contagious a disease is; it provides the average number of people in a susceptible population that a single infected person will spread the disease over the course of their infection [9], and depends on the physical characteristics of coronavirus [13]. The incubation period $\tau$ is the time elapsed between exposure to coronavirus and first symptoms; during this period, an infected individual cannot infect others; other characteristic periods and time intervals are the latent period between exposure and infection, and the generation time, mostly concerned with transmission process [14]. Characteristic values for COVID-19 are $\tau \sim 5–6$ days and $R_0 \sim 2–4$ [15]. The rate expression can be written as $\lambda \propto \ln R_0/\tau$, and involves all the physical characteristics of the mechanisms of infection and the environmental interactions; this can be easily derived, considering difference equations (that is, iterated discrete maps) (e.g., see: [16–18]). Setting the time to be given in discrete $\tau$-steps ($t = 1\tau, 2\tau,...$), then, by definition of $R_0$ (average number of people that a single infected person will spread the disease), we have $N_t = R_0 N_{t-\tau}$, that is,

$$N_t = R_0 N_{t-\tau} = \cdots = R_0^{\frac{t}{\tau}} N_0, \text{ thus,} \tag{3A}$$

$$N_t = N_0 \cdot \exp\left(\frac{1}{\tau}\ln R_0 \cdot t\right). \tag{3B}$$

We note that the number of the infected cases does not vary for times $t$ taken in-between the integer multiples of $\tau$, but this is not expected in mixtures of populations with random characteristics. Indeed, in a mixture of $M$ evolving infected populations with different initial number of cases $N_0^{(m)}$ and starting times $t_0^{(m)}$, $m = 1, 2, \ldots, M$, the total number of infected cases $N$ at a continuous time $t$ is given by

$$N(t) = \sum_{m=1}^{M} N_0^{(m)} \cdot \exp\left[\frac{1}{\tau}\ln R_0 \cdot (t - t_0^{(m)})\right] = N_0 \cdot \exp\left(\frac{1}{\tau}\ln R_0 \cdot t\right), \tag{3C}$$

$$\text{with } N_0 \equiv \sum_{m=1}^{M} N_0^{(m)} \cdot \exp(-\frac{1}{\tau}\ln R_0 \cdot t_0^{(i)}), \tag{3D}$$

which coincides with (3b), but with time $t$ varying on a daily basis, independently of the larger value of $\tau \sim 5$ days [15]. Therefore, we set the readout of the total number of infected cases $N$ on a daily basis, such as: $N_t \equiv N(t/[d])$; then, Eq (3D) matches Eq (2B),

$$N_t = N_0 \exp\left(\frac{1}{\tau}\ln R_0 \cdot t\right) \equiv N_0 \exp(\lambda \cdot t), \tag{4}$$

where the exponential rate is given by:

$$\lambda = (\ln R_0)/\tau. \tag{5}$$

The main factors that can affect the exponential rate $\lambda$ are: (a) culture in social activities, and (b) environmental temperature and/or other thermodynamic parameters. Intense cultural and social activities have reasonably a positive correlation with $R_0$. As previously mentioned, measures against the virus spread do not effectively influence the exponential growth rate; e.g., they do not change the culture in social activities, which are characteristics of the particular population, but they just cease these social activities for some period of time. In terms of modeling, measures appear only in the negative feedback factor $I$ and not in the $E$ factor of model in Eq (1), while the culture, together with the environmental temperature, are the two main parameters affecting $R_0$ directly. Potentially, the environmental temperature $T$ can affect all the parameters influencing exponential rate. We approach this dependence by (i) a linear approximation of the phenomenological relationship between exponential rate and temperature, and (ii) the connection of reproduction number with Arrhenius behavior (with negative activation energy):

(i) The temperature can affect the physical properties of coronavirus, such as, the incubation time $\tau$, as well as, the reproduction number $R_0$ that depends on these physical properties [13]. A linear approximation absorbs the (weak) temperature dependence of any parameters involved in the exponential rate; then, Eq (5) is linearly expanded as:

$$\lambda(T) \cong \frac{1}{\tau}\ln(R_0^{\text{NTP}}) + |\partial\lambda/\partial T|^{\text{NTP}} \cdot (T^{\text{NTP}} - T) + O(T^2), \tag{6}$$

where we set the intercept to be given in normal conditions of atmospheric temperature and

pressure (NTP) (that is, $T = 20 \, C^0$, $P = 1$atm). Then, we rewrite the exponential rate as:

$$\lambda(T) = \lambda_0 \cdot (1 - T_C^{-1} \cdot T), \text{ with} \tag{7A}$$

$$\lambda_0 \equiv \frac{1}{\tau}\ln(R_0^{NTP}) + |\partial\lambda/\partial T|^{NTP} \cdot T^{NTP}, \; p_2 = -\lambda_0 T_C^{-1} \equiv -|\partial\lambda/\partial T|^{NTP}, \tag{7B}$$

where $\lambda_0$ and $p_2$ are the intercept and slope of the linear relation (7a).

(ii) Coronavirus uses their major surface spike protein to bind on a receptor—another protein that acts like a doorway into a human cell [19]. The whole process is a slow chemical reaction, where the mechanism behind can lead to rates negatively correlated with temperature, i.e., increasing rate with decreasing temperature. This is consistent to reaction rate expressed by the Arrhenius exponential with negative activation energy $\exp[|E_a|/(k_B T)]$ [20]. Then, the effective reproduction number $R_0(T)$ is expressed as a product combining the reproduction number in the absence of temperature effect, $R_0^\infty$, and the Arrhenius exponential rate, namely,

$$R_0(T) = R_0^\infty \cdot \exp[|E_a|/(k_B T)], \text{ with } R_0^\infty = R_0^{NTP} \cdot \exp[-|E_a|/(k_B T^{NTP})]. \tag{8}$$

Then, Eq (5) gives

$$\lambda(T) = -\frac{1}{\tau}\left[|E_a|/(k_B T^{NTP}) - \ln(R_0^{NTP})\right] + \frac{1}{\tau}(|E_a|/k_B) \cdot T^{-1}. \tag{9}$$

We rewrite this expression as:

$$\lambda(T) = \lambda_0 \cdot (-1 + T_C \cdot T^{-1}), \text{ with} \tag{10A}$$

$$\lambda_0 \equiv \frac{1}{\tau}[|E_a|/(k_B T^{NTP}) - \ln(R_0^{NTP})], \; p_2 = \lambda_0 T_C \equiv |E_a|/(\tau k_B), \tag{10B}$$

where $-\lambda_0$ and $p_2$ are the intercept and slope of the linear relation (10a), respectively.

Reactions of negative activation energy are barrier-less, relying on the capture of the molecules in a potential well. Increasing {decreasing} the temperature leads to a reduced {gained} probability of the colliding molecules capturing one another. Due to the negative activation energy, decreasing the environmental temperature reduces the probability of virus-protein reaction, thus the virus may stay inactive on air or surfaces and eventually die.

Exponential growth is related to community spread through outdoors activities, while the decelerated growth caused by effective measures is related to indoors activities: Exponential growth exists once the disease is still effective and the measures are loosened, allowing people to outdoor social activities; however, exponential growth decelerates followed by the decay phase, once effective measures hold people in small groups indoors. Therefore, exponential growth rate must be related to outdoors (rather than indoors) activities, and thus to the environmental temperature. As long as the exponential growth takes place, the environmental temperature has an effective role on the chemical reaction between virus and spike protein.

It should be noted that both the models (7b) and (10b) consider that the exponential rate $\lambda$, or the reproduction number $R_0$, are subject to a component influenced by the culture in social activities (intercept $\lambda_0$) and a component mostly influenced by the temperature (linear term with slope $p_2$). In this way, the slope may indicate some universal quantity involved, such as, the (negative) activation energy.

Next, we employ the above two expressions of exponential rate λ and temperature $T$, Eqs (7A and 10A), in order to set the two types of statistical models for fitting $(T, \lambda)$ measurements for US and Italian regions.

## 3. Methods

### 3.1 Data

We use publicly available datasets of: (1) average environmental temperature of US and Italian regions (e.g., see: www.ncdc.noaa.gov/data-access/land-based-station-data/land-based-datasets/climate-normals; it.climate-data.org; www.weather-atlas.com); (2) time series of the number of daily infected cases of US and Italian regions (e.g., see: www.thelancet.com; www.protezionecivile.gov.it).

### 3.2 Data analysis

We analyze the datasets of regional infected cases in US and Italy, derive the relationship of the exponential growth rate of the number of cases with temperature, and evaluate its statistical confidence. First, we derive the exponential growth rates of the infected cases characterizing each examined region of US and Italy; then, we plot these values against the environmental temperatures of each region, and perform the corresponding statistical analysis. We proceed according to the following steps:

i.  Collect the time series of the current infected cases $N_t$ for all US and Italian regions.

ii.  For each of the US and Italian regions, we plot log $(N_t)$ and log $(\Delta N_t)$ with time $t$, detect the time intervals of linear relationship corresponding to the phase of exponential growth, fit the data-points within this region, and derive the slope (on linear-log scale), that is, the exponential growth rate λ. The total $N_t$ and new cases $\Delta N_t$ should be characterized by the same exponential rate, λ, thus the slopes resulted from the linear fits of log $(N_t)$ and log $(\Delta N_t)$ with time are (weighted) averaged (Fig 4).

iii.  Collect environmental temperature data, and calculate the temperature averaged over the whole examined region. The incubation period $\tau$ is longer than the time scale of a single day or night, thus the temperature is averaged over the daily and nightly measurements.

iv.  Co-plot all the derived sample values $(T \pm \delta T, \lambda \pm \delta \lambda)$, where each pair corresponds to each examined region; then, apply a linear fitting in order to derive the linear relationship between $T$ and λ, as well as, evaluate the statistical confidence of this relationship; repeat the same for all US and Italian regions.

v.  Determine the critical temperature $T_C$ for which the rate becomes negligible; to eliminate the uncertainties of $T_C$ as a fitting parameter, we perform the linear fitting with the statistical model $\lambda = \lambda_0 (1 - T/T_C)$ instead of $\lambda = p_1 + p_2 \cdot T$.

vi.  Repeat (iv) and (v) with pairs of $(T^{-1} \pm \delta T^{-1}, \lambda \pm \delta \lambda)$; we estimate again $T_C$ by performing the linear fitting with the statistical model $\lambda = \lambda_0 (-1 + T_C/T)$ instead of $\lambda = p_1 + p_2 \cdot T^{-1}$.

### 3.3 Statistical analysis

The hypothesis to be tested is that the exponential growth rate λ varies linearly with temperature; ($x$ is set to be the temperature or its inverse). This is tested by examining the chi-square corresponding to the fitting of the two-parameter linear statistical model $\lambda(x; p_1, p_2) = p_1 + p_2 x$ to

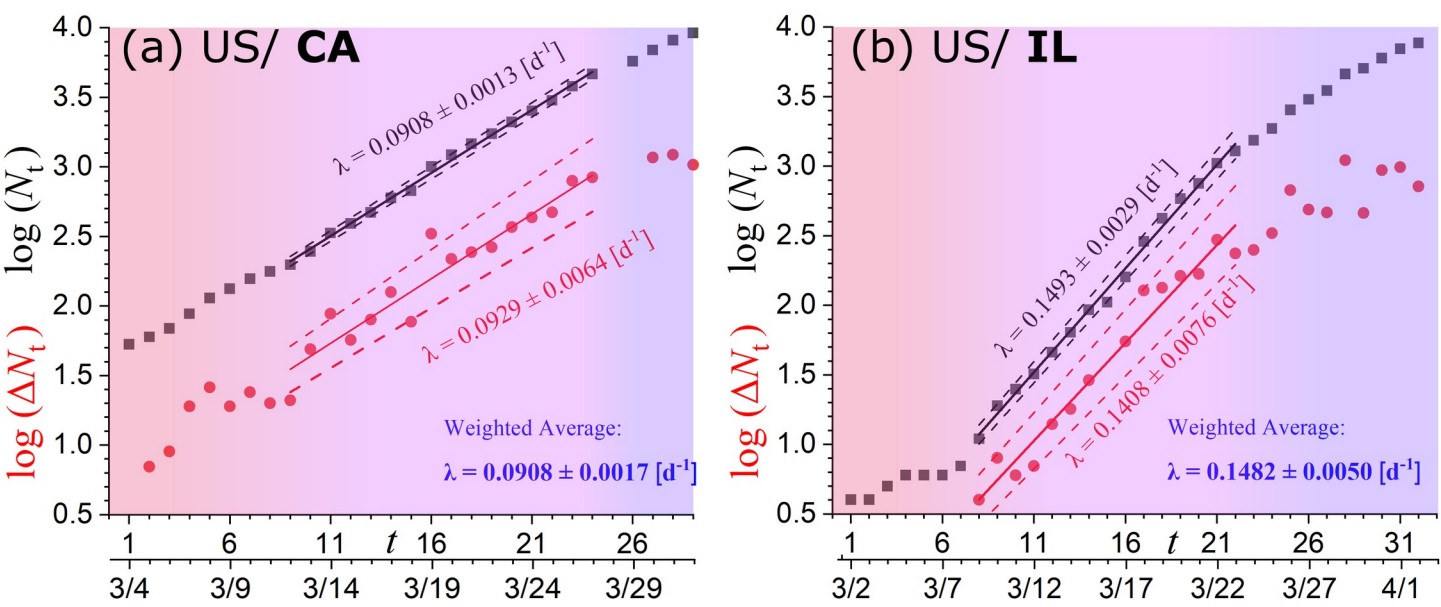

**Fig 4. Linear fitting of the number of the total $N_t$ and new $\Delta N_t$ infected cases with time (on linear-log scale) for the states of California and Illinois, where the slope reads the exponential rate λ.** The resulted rates from the linear fitting of log ($N_t$) (black) and log ($\Delta N_t$) (red) are (weighted) averaged (blue). The phases are color-coded as in Fig 1.

the given $N$ data points, (The number of data point, $N$, should not to be confused with the number of cases, $N_t$). Therefore, we minimize the chi-square $\chi^2(p_1, p_2) = \sum_{i=1}^{N} \sigma_i(p_2)^{-2}$ $(\lambda_i - p_1 - p_2 x_i)^2$, where the total variance that characterize each data point is now given by $\sigma_i(p_2)^2 = \sigma_{\lambda i}^2 + p_2^2 \sigma_{xi}^2$ [21]. The global minimum of the chi-square function $\chi^2(p_1, p_2)$ gives the optimal parameter values, $(p_1^*, p_2^*)$, by solving the normal equations $\partial \chi^2(p_1, p_2)/\partial p_1 = 0$ and $\partial \chi^2(p_1, p_2)/\partial p_2 = 0$; the minimum chi-square value is $\chi_{\min}^2 = \chi^2(p_1^*, p_2^*)$. The statistical errors of these values are given by $\delta p_{\alpha, \text{st}}^* = \sqrt{\chi_{\text{red}}^2 \cdot H_{\alpha\alpha}^{-1}}$, $\alpha = 1,2$, where $H$ is the Hessian matrix of the chi-square at the global minimum, and $H_{\alpha\alpha}^{-1}$ is the $\alpha$-th diagonal element of its inverse matrix [22,23]; $\chi_{\text{red}}^2 = \frac{1}{N-2}\chi_{\min}^2$ is the reduced chi-square value for degrees of freedom (dof) equal to $M = N$-2. The propagation errors of the measurements $\{(x_i \pm \sigma_{xi}, \lambda_i \pm \sigma_{\lambda i})\}_{i=1}^{N}$ are given by

$\delta p_{\alpha, \text{pr}}^* = \sqrt{(\partial p_\alpha^*/\partial x_i)^2 \sigma_{xi}^2 + (\partial p_\alpha^*/\partial \lambda_i)^2 \sigma_{\lambda i}^2}$, $\alpha = 1,2$, where the derivatives are numerically derived.

We will use two linear statistical models, (a) $\lambda(T) = \lambda_0 \cdot (1 - T_C^{-1} \cdot T)$, and (b) $\lambda(T^{-1}) = \lambda_0 \cdot (-1 + T_C \cdot T^{-1})$, as given by (Eqs 7A and 10A); both can be written with the linear expression:

$$\lambda(x; p_1, p_2) = p_1 + p_2 \cdot x, \text{ where} \tag{11A}$$

$$\text{(i) } x = T, \; p_1 = \lambda_0, \; p_2 = -\lambda_0 T_C^{-1}, \text{ and (ii) } x = T^{-1}, \; p_1 = -\lambda_0, \; p_2 = \lambda_0 T_C. \tag{11B}$$

The statistical confidence of the dependence of the exponential growth rate on the environmental average temperature may be sufficiently high, leading to the acceptance of any of the two statistical models. The goodness of the fitting of each model is evaluated using two types of statistical tests, the "reduced chi-square", the "$p$-value of the extremes", and their combination (e.g., [24–26]), while Student's t-test is also used for evaluating the statistical confidence of the derived slopes:

- **Reduced Chi-Square**: The goodness of fitting is estimated by the reduced chi-square value, $\chi^2_{red} = \frac{1}{M}\chi^2_{min}$. The meaning of $\chi^2_{red}$ is the portion of $\chi^2_{min}$ that corresponds to each of the dof, and $\chi^2_{red}$ has to be ~1 for a good fit. Therefore, fitting is characterized as "good" when $\chi^2_{red}$~1, otherwise there is an overestimation, $\chi^2_{red}<1$, or underestimation, $\chi^2_{red}>1$, of the errors. One order of magnitude less, $\chi^2_{red} = 0.1$, or more, $\chi^2_{red} = 10$, can be set as the accepted limits, i.e., $0.1\leq \chi^2_{red}\leq 10$.

- **P-value of the extremes**: The goodness of fitting is evaluated by comparing the estimated minimized chi-square value, $\chi^2_{min}$, and the chi-square distribution,

  $P(\chi^2; M) = \frac{2^{-\frac{M}{2}}}{\Gamma(\frac{M}{2})}e^{-\frac{1}{2}\chi^2}(\chi^2)^{\frac{M}{2}-1}$, that is, the distribution of all the possible $\chi^2$ values (parameterized by the dof = $M$). The likelihood of having a $\chi^2$ value, equal to or larger than the estimated value $\chi^2_{min}$, is given by the complementary cumulative distribution. The probability of taking a result $\chi^2$, larger than the estimated value $\chi^2_{min}$, defines the $p$-value that equals $P(\chi^2_{min} \leq \chi^2 < \infty) = \int_{\chi^2_{min}}^{\infty} P(\chi^2; M)d\chi^2$. The larger the $p$-value, the better the fitting. According to this method, the probability of taking a result with $\chi^2$ being extremer than the observed value $\chi^2_{min}$, defines the $p$-value of the extremes; this equals the minimum between the two probabilities, $P(0 \leq \chi^2 \leq \chi^2_{min})$ and its complementary, $P(\chi^2_{min} \leq \chi^2 < \infty)$. Fits associated with $p$-values smaller than the significance level of 0.05 are typically rejected.

- **Combined P-value and Chi-Square**: The $p$-value of the extremes has very similar behavior with the reduced chi-square [27, 28], because, (i) the $p$-value attains the optimal value ($p = 0.5$) when chi-square does ($\chi^2_{red} = 1$), (ii) larger values of $1$-$2p$ corresponds to larger values of $|\chi^2_{red} - 1|$, (iii) both fractions $(1-2p)/(1+2p)$ and $|1 - \chi^2_{red}|/(1 + \chi^2_{red})$ range from 0 to 1, reduced to 9/11 when the accepted limits, $p = 0.05$ or $0.1\leq\chi^2_{red}\leq10$, are reached. Then, a combined measure can be defined by the sum of the squares of these fractions, i.e.,

  $$\sqrt{\left[(1 - 2p)/(1 + 2p)\right]^2 + \left[(1 - \chi^2_{red})/(1 + \chi^2_{red})\right]^2}.$$

- **Student's t-test**: This is another test for evaluating the statistical confidence of the slope derived from the linear fitting of the temperature-rate sample points $(T_i\pm\delta T_i, \lambda_i\pm\delta\lambda_i)$ and $(T_i^{-1}\pm\delta T_i^{-1}, \lambda_i\pm\delta\lambda_i)$. We examine, whether the slope $p_2\pm\delta p_2$ has significant difference from the zero slope (null hypothesis: slope is zero), by performing the Student's $t$-test with $t_m = p_2/\delta p_2$, where the corresponding $p$-value is derived from the integration of $t$-distribution $P_t(t; M) = \frac{\Gamma[\frac{1}{2}(M+1)]}{\sqrt{\pi M}\Gamma(\frac{M}{2})}(1 + t^2/M)^{-\frac{1}{2}(M+1)}$ for $t\leq t_m<\infty$, i.e., $p_t(t_m; M) = \int_{t_m}^{\infty} P_t(t; M)dt$. The Student's $t$-test is not passed for the null hypothesis that the examined slope equals zero, when the corresponding $p_t$-value is smaller than the acceptable confidence limit of 0.05; then, the null hypothesis is rejected, imposing that the slope has statistically significant difference for zero. In addition, we compare the slopes estimated for US with those estimated for Italian regions, by deriving $t_m = |b_{US} - b_{IT}|/\sqrt{\sigma^2_{US} + \sigma^2_{IT}}$ and $M_m = (\sigma^2_{US} + \sigma^2_{IT})^2/(\sigma^2_{US}/M_{US} + \sigma^2_{IT}/M_{IT})$, and then, finding again $p_t(t_m;M_m)$; the $t$-test is passed for the null hypothesis that the examined slopes are equal, when the corresponding $p_t$-value is larger than 0.05; in this case, the null hypothesis is accepted, thus there is no statistically significant difference between the two slopes.

## 4. Results

The linear fitting of log ($N_t$) or log ($\Delta N_t$) with respect to time $t$ within the region of exponential growth phase, resulted to the respective rates (which are given by the fitted slopes); their weighted averages are shown in Table 1 for US and in Table 2 for Italian regions, while plotted

against the average regional temperature in Figs 5 and 6, respectively. The method of weighted fitting for double uncertainties $(x_i \pm \delta x_i, \lambda_i \pm \delta\lambda_i)$, as described by [21], is used for estimating the fitting parameters $\lambda_0$, $T_C$, together with their statistical, propagation, and total errors. The fits of the linear statistical model with temperature, $x_i = T_i$, (left panels in Figs 5 and 6), as well as of the alternative statistical model with inverse temperature, $x_i = T_i^{-1}$, $\delta x_i = \delta T_i \cdot T_i^{-2}$, (right panels in Figs 5 and 6), are both characterized with high statistical confidence, attaining high $p$-values ($>0.05$) and reduced chi-squares $\chi^2_{red}$ values (close to 1); also, both fits provide similar estimations of $T_C$. The fitting results are shown in Table 3.

We also examine whether the sample points $(T_i \pm \delta T_i, \lambda_i \pm \delta\lambda_i)$ are subject to statistically significant concentrations or rarefactions, namely, whether possible heterogeneities within the distribution of sample points plays significant role in the fitted relationship. For this, we derive the temperature-rate relationship and its statistical confidence by fitting the homogenized set of sample points, instead of the raw sample points; then, we examine whether the fitting parameters differ from those derived from fitting the raw sample points. We homogenize the sample points by grouping them in temperature bins of $\Delta T \sim 1 \ C^0$ (e.g., see: [29]). We estimate the weighted mean and error of the rates included in each bin. In the case of US regions we

**Table 1. Averaged temperatures and estimated exponential rates of US regions.**

| Region | $T$ [C$^0$] | $\delta T$ [C$^0$] | $\lambda$ [d$^{-1}$] | $\delta\lambda$ [d$^{-1}$] |
|---|---|---|---|---|
| MI | -1.23 | 4.2 | 0.1709 | 0.0096 |
| WI | -0.7 | 2.6 | 0.1704 | 0.0099 |
| MA | 1.7 | 3.5 | 0.1495 | 0.0183 |
| CT | 2.9 | 4.0 | 0.1463 | 0.0062 |
| PA | 3.1 | 2.9 | 0.1349 | 0.0019 |
| WA | 3.7 | 3.5 | 0.1432 | 0.0162 |
| NJ | 4.5 | 3.9 | 0.1523 | 0.0114 |
| IN | 5.1 | 5.2 | 0.1237 | 0.0064 |
| OH | 5.1 | 4.2 | 0.1256 | 0.0092 |
| IL | 5.2 | 4.0 | 0.1482 | 0.0050 |
| CO | 5.6 | 4.6 | 0.1108 | 0.0200 |
| NY | 5.8 | 2.8 | 0.1203 | 0.0096 |
| MO | 6.5 | 2.8 | 0.1322 | 0.0079 |
| VA | 7.4 | 2.8 | 0.0946 | 0.0015 |
| TN | 8.8 | 5.2 | 0.1266 | 0.0098 |
| NC | 9.7 | 1.9 | 0.1398 | 0.0073 |
| SC | 11.3 | 2.8 | 0.1027 | 0.0044 |
| GA | 13.3 | 2.8 | 0.1169 | 0.0096 |
| CA | 14.2 | 3.9 | 0.0908 | 0.0017 |
| LA | 15.2 | 4.2 | 0.1081 | 0.0122 |
| TX | 15.3 | 5.7 | 0.1083 | 0.0070 |
| AZ | 17.0 | 2.7 | 0.0786 | 0.0087 |
| FL | 19.5 | 4.6 | 0.1033 | 0.0072 |

(1) The given values of exponential growth rate and its uncertainty is the result of the weighted averaging of the rates derived from total and new infected cases; (2) the environmental temperature is averaged over the time period, from $\tau \sim 5$ days before the appearance of the 1st case, to 1st April; (3) the standard deviation of temperature is given by the half difference between highest and lowest values within the examined time period, divided by $\sqrt{2}$ (similar to the standard deviation for a sinusoidal function); (4) NY: The temperature refers to the New York City, instead of the whole state, which suffers from the vast majority of the state infected cases.

**Table 2. Averaged temperatures and estimated exponential rates of Italian regions.**

| Region | $T$ | $\delta T$ | $\lambda$ | $\delta\lambda$ |
|---|---|---|---|---|
| Aosta Valley | -4.0 | 2.8 | 0.1187 | 0.0041 |
| S Tyrol | 2.2 | 2.7 | 0.0887 | 0.0080 |
| Abruzzo | 4.8 | 3.1 | 0.1055 | 0.0127 |
| Piedmont | 6.2 | 3.1 | 0.0983 | 0.0068 |
| Molise | 7.5 | 3.2 | 0.0667 | 0.0158 |
| Basilicata | 7.5 | 3.2 | 0.0621 | 0.0141 |
| Friuli Venezia Giulia | 7.5 | 1.8 | 0.0816 | 0.0147 |
| Veneto | 7.6 | 2.5 | 0.0674 | 0.0027 |
| Liguria | 7.7 | 2.4 | 0.0693 | 0.0106 |
| Tuscany | 7.8 | 3.0 | 0.0900 | 0.0127 |
| Umbria | 8.1 | 3.1 | 0.0900 | 0.0127 |
| Lazio | 11.1 | 2.4 | 0.0906 | 0.0065 |
| Lombardy | 8.1 | 2.9 | 0.0942 | 0.0089 |
| Emilia Romagna | 8.3 | 3.0 | 0.0695 | 0.0095 |
| Marche | 10.4 | 2.0 | 0.0696 | 0.0140 |
| Calabria | 10.5 | 2.5 | 0.0939 | 0.0155 |
| Sicily | 11.0 | 2.3 | 0.0762 | 0.0053 |
| Campania | 11.6 | 2.0 | 0.0604 | 0.0076 |

Similar to the notes (1)-(3) of Table 1.

also performed a homogenization of rates, by grouping the temperature-binned means in rate bins of $\Delta\lambda \sim 0.01$ d$^{-1}$. In the case of sample points with inverse temperatures, $(T_i^{-1} \pm \delta T_i^{-1}, \lambda_i \pm \delta\lambda_i)$, the procedure is exactly the same. Homogenized datasets result in a smooth relationship between the values of binned temperature and rate, as it can be observed in the plots of rate against temperature or inverse temperature (left or right lower panels, respectively), and for both US and Italy regions (Figs 5 and 6, respectively). The results are highly supportive of the negative correlation between rate and temperature. The results are shown in Table 4. We observe that the linear relationships of the growth rate with temperature or inverse temperature are characterized by high statistical confidence for the homogenized datasets ($p$-values much higher than the significant limit of 0.05; $\chi^2_{\text{red}}$ far from the significant limits of 0.1 and 10). Therefore, the arrangement of sample points do not affect significantly the fitting results.

In addition, as shown in Tables 3 and 4, the linear fits of sample points $(T_i \pm \delta T_i, \lambda_i \pm \delta\lambda_i)$ and $(T_i^{-1} \pm \delta T_i^{-1}, \lambda_i \pm \delta\lambda_i)$ do not pass the Student's $t$-test for the null hypothesis that their slopes equals zero, i.e., the corresponding $p_t$-values are smaller than the acceptable confidence limit of 0.05; therefore, the negative correlation of environmental temperature with the exponential rate is statistically significant (accepted with confidence 95%).

In order to improve the statistics of the estimated critical temperature, we combine the sample points $(T_i \pm \delta T_i, \lambda_i \pm \delta\lambda_i)$ of US and Italian regions. First, we perform the Student's $t$-test to compare the slopes from these regions; we find high $p_t$-values ($>0.05$) for both fits of $x = T$ and $x = T^{-1}$, thus, the two populations are likely characterized by the same slope. The respective intercept $\lambda_0$ does not pass the same test, i.e., the intercepts corresponding to US and Italian regions are likely different; (that is expected, given of the different culture). A universality may characterize the slopes of two countries, either for the fits with $x = T$ or $x = T^{-1}$, i.e., $p_2 = (\partial\lambda/\partial T)^{\text{NTP}}$ or $p_2 = |E_a|/(\tau k_B)$, respectively.

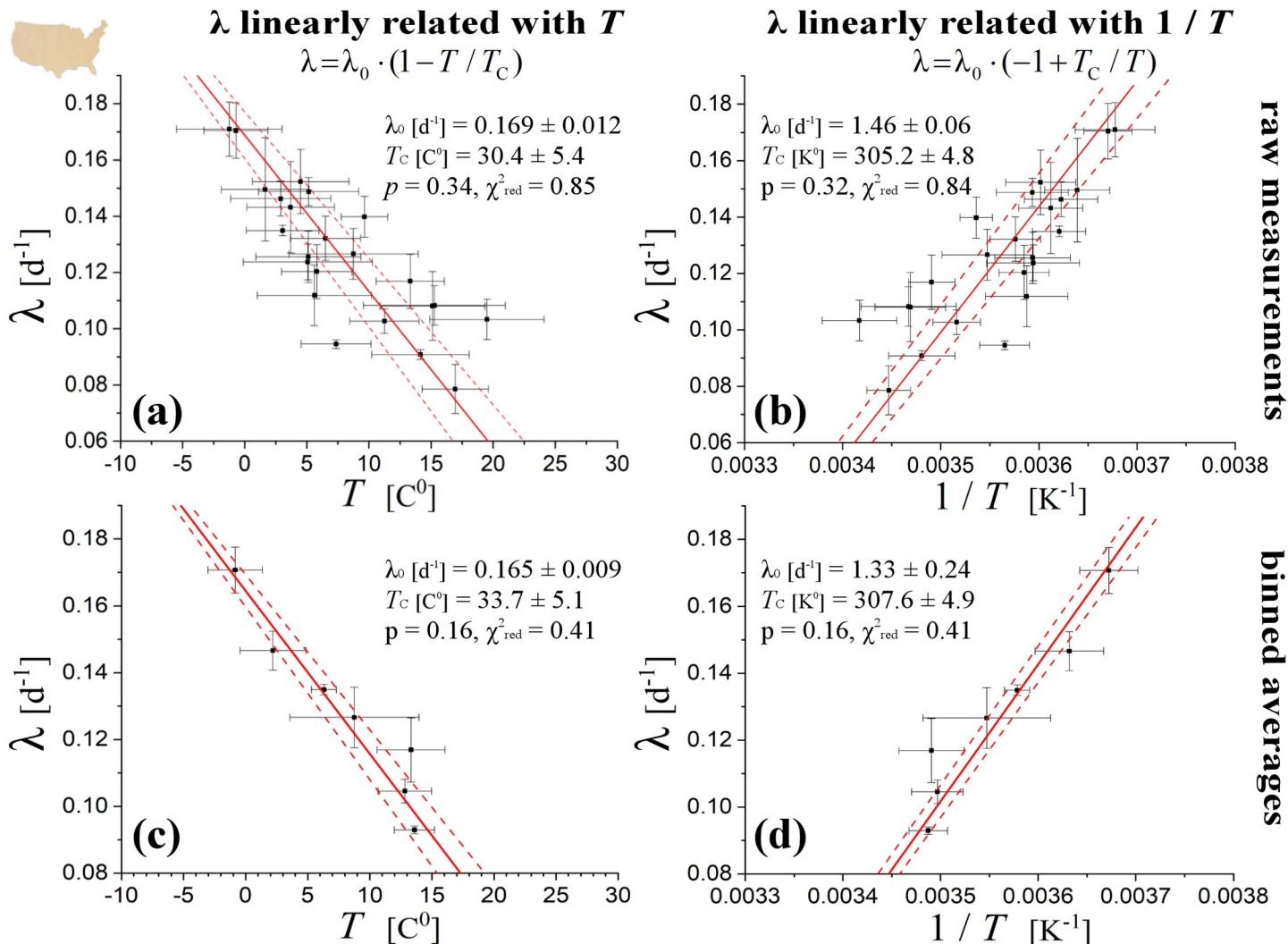

**Fig 5. Linear fitting of rates with (left) temperatures and (right) inverse temperatures for US regions.** The fitting is weighted with double uncertainties (i.e., on both the temperature and rate values). The analysis is first completed for the raw measurements (upper) and then repeated for the binned averages (lower).

Next, we perform the linear fits of the sample points $(T_i \pm \delta T_i, \lambda_i \pm \delta \lambda_i)$ and $(T_i^{-1} \pm \delta T_i^{-1}, \lambda_i \pm \delta \lambda_i)$ for the mixed set of US and Italian data, once the rates of the Italian regions are shifted by $\Delta\lambda$; (this is allowed, since it has just be shown that a universality is likely characterized the slopes). The optimal fitting is obtained for that shift $\Delta\lambda$, for which the reduced chi-square is ~1, the *p*-value of the extremes is ~0.5, and the combined measure ~0 (see previous section). Fig 7 shows how the combined datasets of temperature-rates from US and Italian regions lead to the optimal fitting. (Note that the optimization is not performed for the binned datasets, since they are characterized by smaller *p*-values–see, Figs 5 and 6). The results are shown in Table 5; we observe that the optimization is reached for two values of the shift $\Delta\lambda$; we estimate the weighted average of the results corresponding to the two shifts. The weighted mean is performed separately for the fitting cases of $x = T$ and $x = T^{-1}$; however, the weighted mean of the critical temperature is performed for all four results.

Table 5 includes the weighted means of slopes for the fits $x = T$ or $x = T^{-1}$, with slopes $p_2 = -|\partial\lambda/\partial T|^{\mathrm{NTP}}$ and $p_2 = |E_a|/(\tau k_\mathrm{B})$, respectively. The latter can be used for deriving the activation

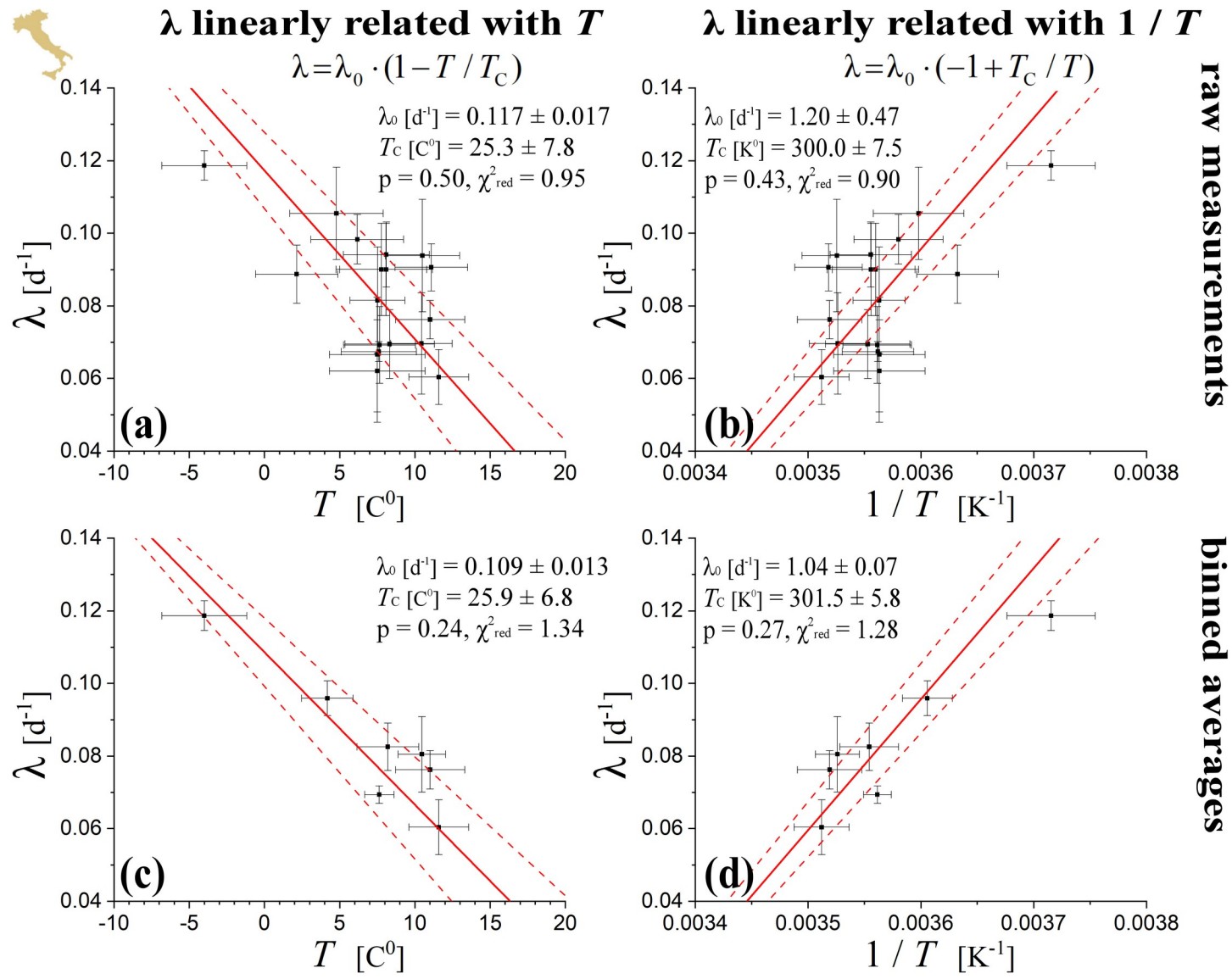

**Fig 6. As in Fig 5, but for Italian regions.**

energy, where $\tau = 5.2 \pm 1.1$ d [15], where we find $|E_a| = 0.212 \pm 0.057$ eV. The value of the critical temperature is $T_C = 303.2 \pm 2.4$ K ($86.1 \pm 4.3$ F$^0$ or $30.1 \pm 2.4$ C$^0$). In addition, using (Eqs 7B and

**Table 3. Fitting parameters of temperature–rate values for US and Italian regions.**

| US/ Model | $\lambda_0$ [d$^{-1}$] | $\delta\lambda_0$ [d$^{-1}$] | $p_2$ * | $\delta p_2$ | $T_C$ [C$^0$] | $\delta T_C$ [C$^0$] | $\chi^2_{red}$ | $p$-value | $p_t$-value |
|---|---|---|---|---|---|---|---|---|---|
| $x = T, p_1 = \lambda_0, p_2 = -\lambda_0 T_C^{-1}$ | 0.1688 | 0.0125 | -0.00554 | 0.00133 | 30.4 | 5.4 | 0.85 | 0.34 | $2.2 \times 10^{-4}$ |
| $x = T^{-1}, p_1 = -\lambda_0, p_2 = \lambda_0 T_C$ | 1.455 | 0.288 | 444 | 107 | 32.0 | 4.8 | 0.84 | 0.32 | $2.3 \times 10^{-4}$ |
| Italy/ Model | $\lambda_0$ [d$^{-1}$] | $\delta\lambda_0$ [d$^{-1}$] | $p_2$ | $\delta p_2$ | $T_C$ [C$^0$] | $\delta T_C$ [C$^0$] | $\chi^2_{red}$ | $p$-value | $p_t$-value |
| $x = T, p_1 = \lambda_0, p_2 = -\lambda_0 T_C^{-1}$ | 0.1173 | 0.0169 | -0.00465 | 0.00206 | 25.3 | 7.8 | 0.95 | 0.49 | 0.019 |
| $x = T^{-1}, p_1 = -\lambda_0, p_2 = \lambda_0 T_C$ | 1.20 | 0.47 | 361 | 166 | 26.7 | 7.5 | 0.90 | 0.43 | 0.022 |

* Units of the slope $p_2$ are [d$^{-1}$K$^{-1}$] when $x = T$, and [d$^{-1}$K] $x = T^{-1}$.

**Table 4. Fitting parameters of binned temperature–rate values for US and Italian regions.**

| US/ Model | $\lambda_0$ [d$^{-1}$] | $\delta\lambda_0$ [d$^{-1}$] | $p_2$ * | $\delta p_2$ | $T_C$ [C$^0$] | $\delta T_C$ [C$^0$] | $\chi^2_{red}$ | $p$-value | $p_t$-value |
|---|---|---|---|---|---|---|---|---|---|
| $x = T, p_1 = \lambda_0, p_2 = -\lambda_0 T_C^{-1}$ | 0.1645 | 0.0089 | -0.00488 | 0.00096 | 33.7 | 5.1 | 0.41 | 0.16 | $1.9\times10^{-3}$ |
| $x = T^{-1}, p_1 = -\lambda_0, p_2 = \lambda_0 T_C$ | 1.331 | 0.2417 | 409 | 81 | 34.4 | 4.9 | 0.41 | 0.16 | 0.021 |
| Italy/ Model | $\lambda_0$ [d$^{-1}$] | $\delta\lambda_0$ [d$^{-1}$] | $p_2$ | $\delta p_2$ | $T_C$ [C$^0$] | $\delta T_C$ [C$^0$] | $\chi^2_{red}$ | $p$-value | $p_t$-value |
| $x = T, p_1 = \lambda_0, p_2 = -\lambda_0 T_C^{-1}$ | 0.1086 | 0.0127 | -0.00420 | 0.00154 | 25.9 | 6.8 | 1.34 | 0.24 | $2.0\times10^{-3}$ |
| $x = T^{-1}, p_1 = -\lambda_0, p_2 = \lambda_0 T_C$ | 1.0374 | 0.2889 | 313 | 115 | 28.3 | 5.8 | 1.28 | 0.27 | 0.021 |

* Same units as in Table 3.

19B) we derive the reproduction number $R_0$, i.e.,

$$\frac{1}{\tau}\ln(R_0^{\mathrm{NTP}}) = \lambda_0^{x=T} - |p_2^{x=T}| \cdot (T^{\mathrm{NTP}}/[\mathrm{C}^0]), \quad \frac{1}{\tau}\ln(R_0^{\mathrm{NTP}}) = p_2^{x=T^{-1}} \cdot (T^{\mathrm{NTP}}/[\mathrm{K}])^{-1} - \lambda_0^{x=T^{-1}}, \text{ or}(12\mathrm{A})$$

$$\frac{1}{\tau}\ln(R_0^{\mathrm{NTP}}) = \lambda_0^{x=T} \cdot (1 - T^{\mathrm{NTP}} \cdot T_C^{-1}), \quad \frac{1}{\tau}\ln(R_0^{\mathrm{NTP}}) = \lambda_0^{x=T^{-1}} \cdot (T_C/T^{\mathrm{NTP}} - 1). \quad (12\mathrm{B})$$

The two formulae in Eq (12B) provide the value of $\frac{1}{\tau}\ln(R_0^{\mathrm{NTP}})$ as 0.0572±0.0098 and 0.0534 ±0.0146, respectively, with weighted mean 0.0560±0.0084; then, we find $R_0^{\mathrm{NTP}} \cong 1.34 \pm 0.10$, that is, the reproduction number for $T$ = 20 C$^0$. The corresponding number at $T$ = 0 C$^0$ is $R_0(0\mathrm{C}^0) \cong 2.47 \pm 0.45$, while by substituting the estimated parameters in Eq (8), we derive the

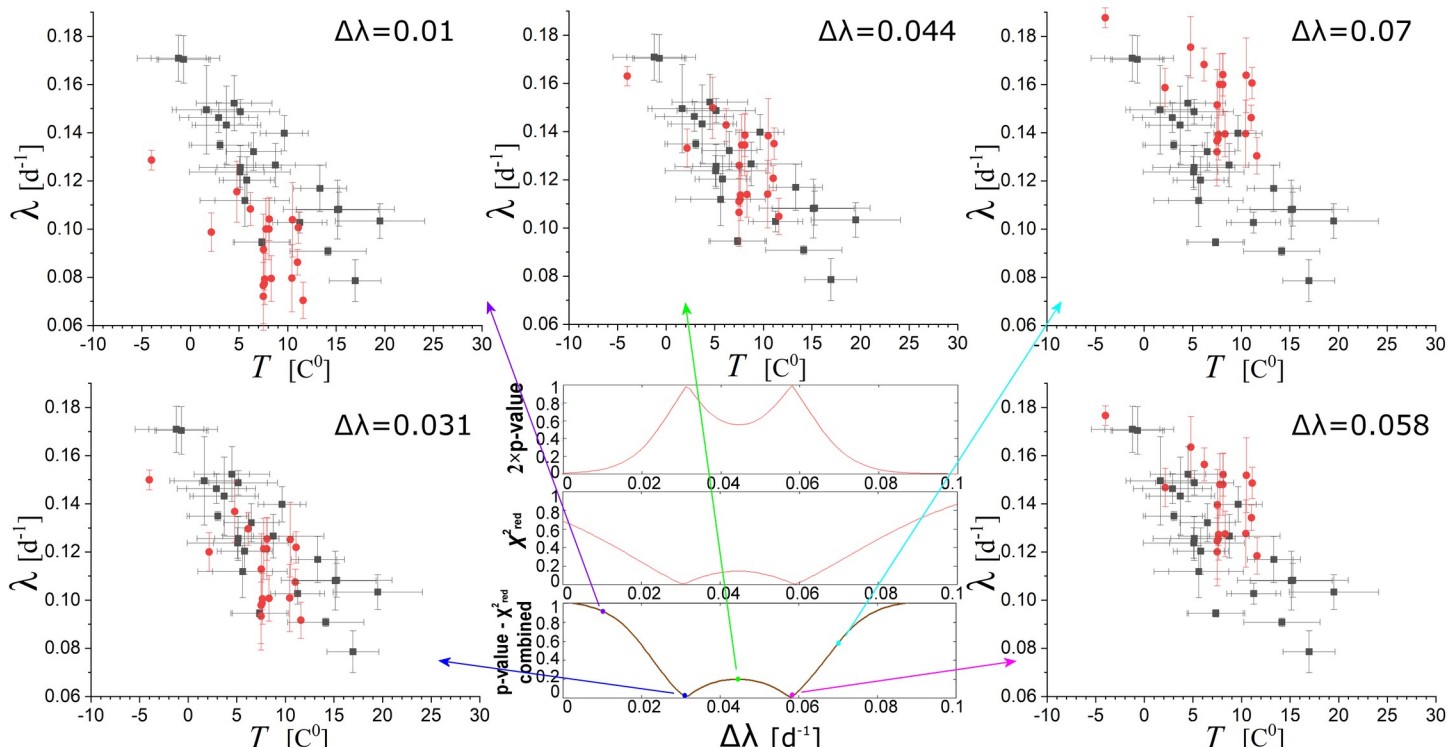

**Fig 7. Fitting of datasets combined for US and Italian regions, with the latter's rates shifted by Δλ.** The optimal fitting corresponds to shifts Δλ~ 0.031 and ~0.058, for which the reduced chi-square is ~1, the $p$-value of the extremes is ~0.5, and the combined measure ~0.

**Table 5. Fitting parameters of combined US and optimally shifted Italian regions.**

| $\Delta\lambda$ [d⁻¹], x = T | $\lambda_0$ [d⁻¹] | $\delta\lambda_0$ [d⁻¹] | $p_2$ [d⁻¹K⁻¹] | $\delta p_2$ [d⁻¹K⁻¹] | $T_C$ [C⁰] | $\delta T_C$ [C⁰] | $T_C$ [F⁰] | $\delta T_C$ [F⁰] |
|---|---|---|---|---|---|---|---|---|
| 0.03098 | 0.1631 | 0.0116 | 0.005717 | 0.00136 | 28.538 | 4.97058 | 83.3684 | 8.947044 |
| 0.05766 | 0.1781 | 0.0113 | 0.005853 | 0.00126 | 30.429 | 4.85595 | 86.7722 | 8.74071 |
| Weighted Mean | 0.1708 | 0.0110 | 0.00579 | 0.00093 | - | - | - | - |
| $\Delta\lambda$ [d⁻¹], x = T⁻¹ | $\lambda_0$ [d⁻¹] | $\delta\lambda_0$ [d⁻¹] | $p_2$ [d⁻¹K] | $\delta p_2$ [d⁻¹K] | $T_C$ [C⁰] | $\delta T_C$ [C⁰] | $T_C$ [F⁰] | $\delta T_C$ [F⁰] |
| 0.02926 | 1.531 | 0.303 | 463 | 113 | 29.4 | 4.3 | 84.9 | 7.8 |
| 0.05899 | 1.575 | 0.282 | 480 | 107 | 31.5 | 4.2 | 88.8 | 7.6 |
| Weighted Mean | 1.555 | 0.208 | 472 | 78 | 30.1* | 2.4 | 86.1 | 4.3 |

* The weighted mean of $T_C$ in [C⁰] or [F⁰] takes into account all four estimated values.

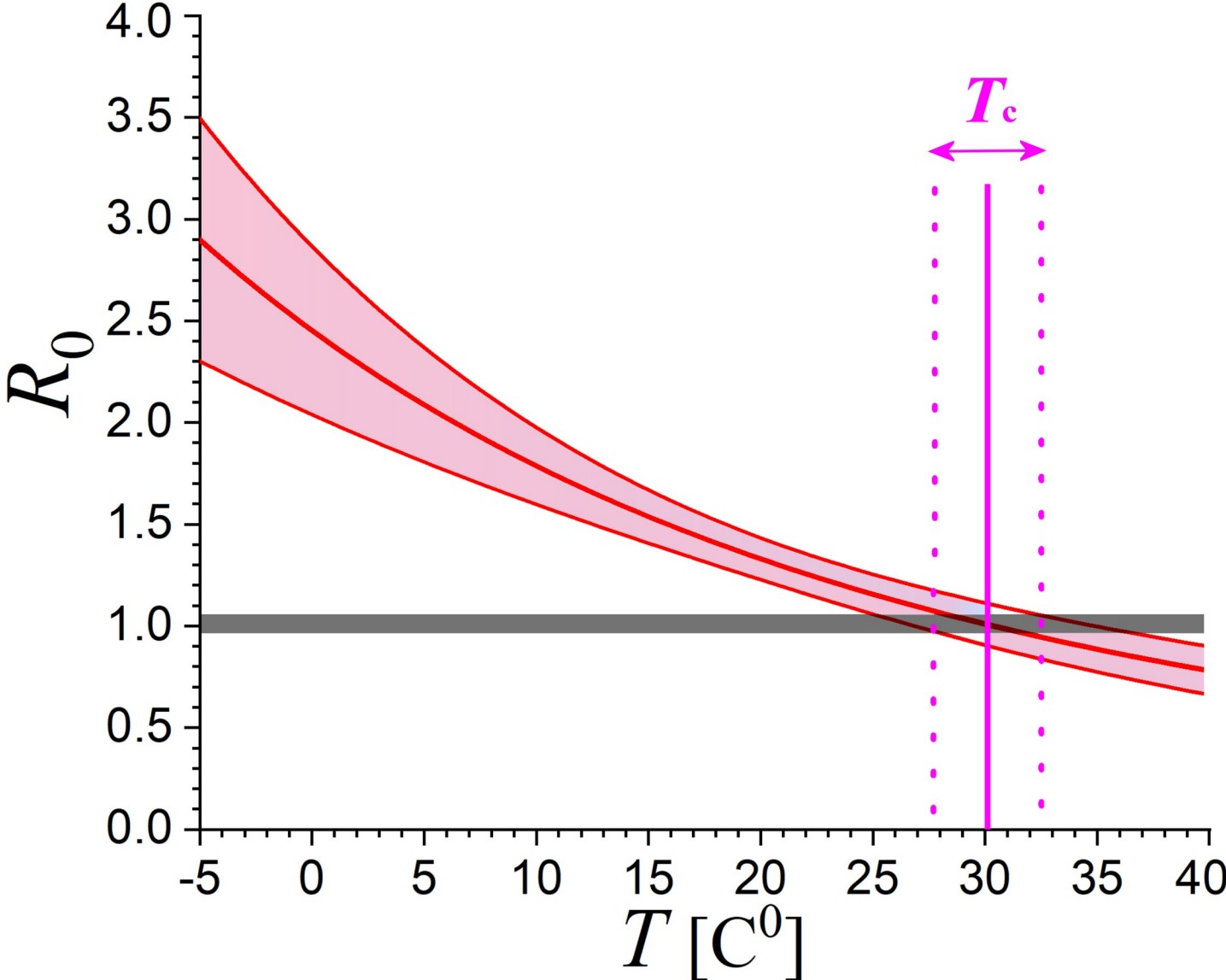

**Fig 8. Relationship of the reproduction number $R_0$ and its uncertainty with environmental temperature T.** According to this, new affected cases cease ($R_0 = 1$) when temperature climbs to $T_C$~30 C⁰ or (~86 F⁰).

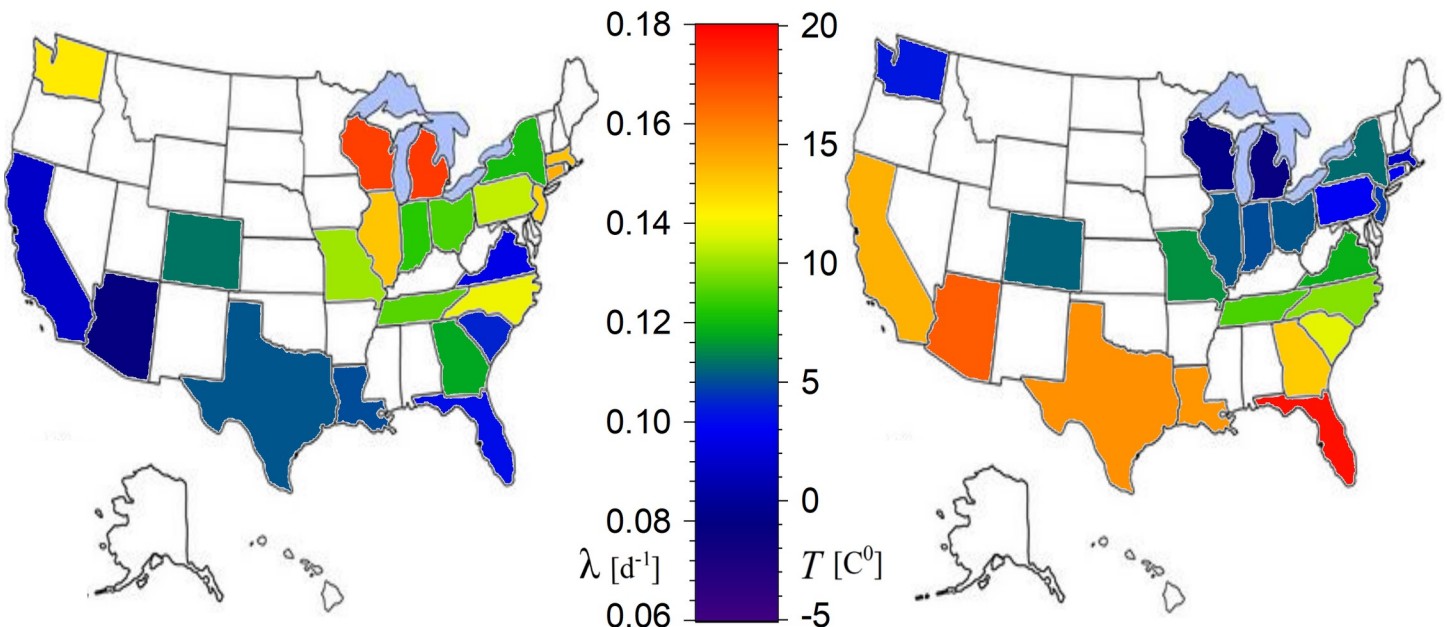

**Fig 9.** Anti-correlation between the spatial distributions of the exponential growth rates of the infected cases (left) and of the average environmental temperature (right).

general relationship for any temperature (expressed in K), also plotted in Fig 8:

$$R_0(T[\text{C}^0]) = (1.334 \pm 0.10) \cdot \exp[(2450 \pm 660)(T^{-1} - 293.15^{-1})]. \tag{13}$$

## 5. Discussion and conclusions

Up-to-date there is no systematic statistical analysis of the effect of the environmental temperature $T$ (and possibly other weather parameters) on the exponential growth rate of the cases infected by COVID-19, while a statistically confident relationship between temperature and growth rate (either with positive or negative correlation) was unknown.

The presented analysis led to the first statistically confident relationship of negative correlation between the exponential growth rate and the average environmental temperature, derived for US and Italian regions. In particular, we analyzed datasets of regional infected cases in US and Italy, derived the exponential growth rates for each of these regions and plotted them against environmental temperatures averaged within the same regions, derived the relationship of temperature—growth rate, and evaluated its statistical confidence.

The performed statistical analysis involved fitting of linear statistical models with the datasets of environmental temperature (or its inverse) and exponential growth rate. The two linear models developed and used for the statistical analysis are (a) $\lambda(T) = \lambda_0 \cdot (1 - T_C^{-1} \cdot T)$, and (b) $\lambda(T^{-1}) = \lambda_0 \cdot (-1 + T_C \cdot T^{-1})$. The statistical confidence of fitting was evaluated using the reduced chi-square values, the $p$-value of extremes, and a testing measure that combines both of these values; also, the Student's $t$-test was used to compare the derived slopes.

The sample points of temperature (or inverse temperature) and exponential growth rate were also tested for statistically significant concentrations or rarefactions, that is, for possible heterogeneities within the distribution of sample points that could have significant role in the results. The statistical analysis of the homogenized temperature-rate data points concluded that the negative correlation between temperature and exponential rate is stable, having no statistically significant variability due to concentrations or rarefactions, and it is characterized by a high statistical confidence.

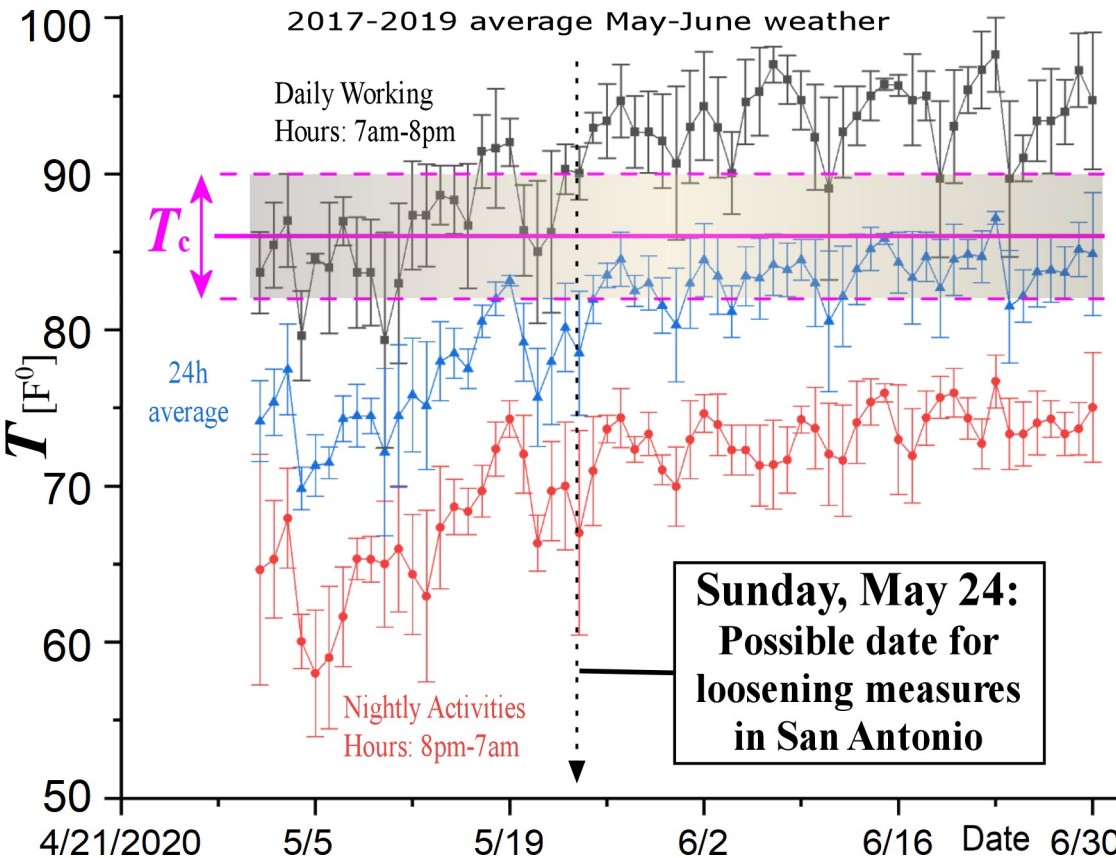

**Fig 10. According to the statistically confident relationship between exponential growth rate of infected cases shown in Fig 8, the critical temperature, which eliminates the exponential growth, and thus the COVID-19 spread, is $T_C = 86.1 \pm 4.3$ F$^0$.** The plot shows also the May-June daily, nightly, and 24h-averaged environmental temperatures in San Antonio, Texas, averaged over the last three years. The daily average temperatures will be clearly above the estimated $T_C$ threshold in the second half of May; thus, the plot suggests a possible date for loosening the strict measures in San Antonio, that is, May 24.

We also performed a Student's *t*-test and ensured that the difference between the sample means of US and Italian regions is not statistically significant. A universality is likely characterizing the slope of the temperature-rate relationship. This verifies the modeling developed and used by this analysis, where the exponential rate λ, or the reproduction number $R_0$, are subject to a component influenced by the culture in social activities (intercept $\lambda_0$) and a component influenced by the temperature (slope $p_2$). In this way, the slope may indicate to a universal quantity involved, such as, the (negative) activation energy.

Having shown that the derived slopes for US and Italian regions are characterized by no statistically confident difference, we improved the statistics of the estimated fitting parameters by combined the sample points of US and Italian regions. From the derived relationship, among others, we were able to estimate the values of the (negative) activation energy $E_a$, as well as the reproduction number $R_0$ at normal conditions and how this depends on temperature.

Therefore, the results clearly showed that there is indeed statistically significant negative correlation of temperature on the exponential growth rate of the cases infected by COVID-19. Fig 9 shows the anti-correlation between the mapped exponential rates and average environmental temperature of the US regions examined by this analysis, which they are characterized by a readable exponential growth phase in their evolution spread curve.

Given the negative correlation of the environmental temperature with the exponential growth rate, it was reasonable to ask for the critical temperature that eliminates the exponential rate, and thus the number of daily new cases in infected regions. This was found to be $T_C \sim 86.1 \pm 4.3$ F$^0$ for US regions. It is straightforward to ask when the environmental temperature will climb above this critical value. As an example, Fig 10 plots the daily average temperatures in San Antonio, Texas, shown that it will be clearly above the estimated $T_C$ threshold by the end of May.

The resulted high statistical confidence of the negative correlation of the environmental temperature on the exponential growth rate of the cases infected by COVID-19 is certainly encouraging for loosening super-strict social-distancing measures, at least, during the summery high temperatures. However, we are, by no-means, recommending a return-to-work date based only on this study. But we do think that this should be part of the decision, as well as an inspiration for repeating the same analysis in other heavily infected regions. The steps of these analyses may be followed as:

i. Identify different outbreaks in regions with the same culture in social activities and different environmental temperature;

ii. Estimate the exponential growth rates for these regions from the time series of infected cases;

iii. Plot the derived rates against the environmental temperature averaged for these regions, and repeat the analysis of this study to determine the temperature-rate relationship and its statistical confidence.

## Author Contributions

**Conceptualization:** George Livadiotis.

**Data curation:** George Livadiotis.

**Formal analysis:** George Livadiotis.

**Funding acquisition:** George Livadiotis.

**Investigation:** George Livadiotis.

**Methodology:** George Livadiotis.

**Project administration:** George Livadiotis.

**Resources:** George Livadiotis.

**Software:** George Livadiotis.

**Supervision:** George Livadiotis.

**Validation:** George Livadiotis.

**Visualization:** George Livadiotis.

**Writing – original draft:** George Livadiotis.

**Writing – review & editing:** George Livadiotis.

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
