## [Decision Letter · Decision Letter 0]

12 May 2020

PONE-D-20-11474

Statistical analysis of the impact of environmental temperature on the exponential growth rate of cases infected by COVID-19

PLOS ONE

Dear Dr Livadiotis,

Thank you for submitting your manuscript to PLOS ONE. After careful consideration, we feel that it has merit but does not fully meet PLOS ONE’s publication criteria as it currently stands. Therefore, we invite you to submit a revised version of the manuscript that addresses the points raised during the review process.

Both reviewers are quite positive about the work, but there are some issues raised by them that have to be addressed before the paper can be accepted.

We would appreciate receiving your revised manuscript by Jun 26 2020 11:59PM. To enhance the reproducibility of your results, we recommend that if applicable you deposit your laboratory protocols in protocols.io, where a protocol can be assigned its own identifier (DOI) such that it can be cited independently in the future. For instructions see: http://journals.plos.org/plosone/s/submission-guidelines#loc-laboratory-protocols

We look forward to receiving your revised manuscript.

Kind regards,

Oscar Millet

Academic Editor

PLOS ONE

Journal Requirements:

1. We note that [Figure(s) 2] in your submission contain [map/satellite] images which may be copyrighted. All PLOS content is published under the Creative Commons Attribution License (CC BY 4.0), which means that the manuscript, images, and Supporting Information files will be freely available online, and any third party is permitted to access, download, copy, distribute, and use these materials in any way, even commercially, with proper attribution. For these reasons, we cannot publish previously copyrighted maps or satellite images created using proprietary data, such as Google software (Google Maps, Street View, and Earth). For more information, see our copyright guidelines: http://journals.plos.org/plosone/s/licenses-and-copyright.

1.    You may seek permission from the original copyright holder of Figure(s) [2] to publish the content specifically under the CC BY 4.0 license. 

Reviewers' comments:

Reviewer's Responses to Questions

**Comments to the Author**

1. Is the manuscript technically sound, and do the data support the conclusions?

Reviewer #1: Yes

Reviewer #2: Yes

2. Has the statistical analysis been performed appropriately and rigorously? 

Reviewer #1: Yes

Reviewer #2: Yes

3. Have the authors made all data underlying the findings in their manuscript fully available?

Reviewer #1: Yes

Reviewer #2: Yes

4. Is the manuscript presented in an intelligible fashion and written in standard English?

Reviewer #1: Yes

Reviewer #2: Yes

5. Review Comments to the Author

Reviewer #1: Dear Editor,

I have read and reviewed the paper “Statistical analysis of the impact of environmental temperature on the exponential growth rate of cases infected by COVID-19” by G. Livadiotis.

The original research of this paper demonstrates the negative correlation between the exponential growth rate of the cases infected by COVID-19 for US and Italian regions and the environmental temperature.

The author uses valid statistical methods to establish an accurate anti-correlation of the infected cases and the environmental temperature. Numerous statistical tests validate the results.

The study derives the critical temperature which may eliminate the daily infected cases. This is a purely statistical approach to such an important matter, and with extra caution, the results could be very useful for future decisions regarding measures against the pandemic.

I therefore recommend the publication of the manuscript after the minor changes I list below:

1. Theory Section, Equation 1.: I recommend that you have the relevant citations before introducing the Equation.

2. Theory Section, 4th paragraph: In the specific sentence I believe the main point is shown by both panels in Figure 3. Wouldn’t also a semi-plot log make this point even more clear?

3. Theory Section, after Equation 5, λ depends on Ro, which could potentially depend on social activities. On the other hand, 4th paragraph of introduction states that the measures do not affect the exponential rate, indicating that there is no potential dependence of parameter λ on Ro. In other words, by reading the introduction I would expect that measures appear only in the negative feedback factor I, and not in the E factor of model in equation (1). I think this point can get a bit clearer either in the introduction or in the theory because it seems contradictive from the first read.

4. Methodology section, point ii): How does the author detects the time interval corresponding to the exponential growth?

5. Results section, paragraph 4: There is a typo in the second to last sentence (“Tthe” should be “The”)

Reviewer #2: There are several awkward expressions and misusage of certain English words. Here are few of them.

1. The statement in line 6 in the abstract is in contradiction of the last sentence and should be corrected: (there is a positive correlation between the average temperature and ...)

2. In the introduction, the word "decay" should be replaced by "decline". In line 5, the word outburst is used. as a verb which it is not and should be replaced by "explode"

3. line 5 in Section 2.2, will spread the disease to over: should read will spread the disease over

3. In section 2.1, The differential equation is written in a non-traditional way x^._t instead of x^.(t). The subscripts are usually reserved for difference equations. The author. needs to explain clearly how he came up with this simple model and describe clearly the functions E and I.

4. The author needs t o explain better the relationship between lambda and R_0 and how he got the equations N_(t+1)=R_0 N_t

5. The author needs to give a better explanation of the derivations of equations 6-10. The author needs to elaborate on the criticality of Arrhenius equation in modeling the effect of temperature on the corona virus.

6. PLOS authors have the option to publish the peer review history of their article (what does this mean?). If published, this will include your full peer review and any attached files.

Reviewer #1: No

Reviewer #2: Yes: Saber Elaydi

---

## [Author Response · Author response to Decision Letter 0]

14 May 2020

Please, see attached file "Response to Reviewers".

---

## [Editor Report · Decision Letter 1]

15 May 2020

Statistical analysis of the impact of environmental temperature on the exponential growth rate of cases infected by COVID-19

PONE-D-20-11474R1

Dear Dr. Livadiotis,

We are pleased to inform you that your manuscript has been judged scientifically suitable for publication and will be formally accepted for publication once it complies with all outstanding technical requirements.

With kind regards,

Oscar Millet

Academic Editor

PLOS ONE
---

## [Editor Report · Acceptance letter]

21 May 2020

PONE-D-20-11474R1 

Statistical analysis of the impact of environmental temperature on the exponential growth rate of cases infected by COVID-19 

Dear Dr. Livadiotis:

I am pleased to inform you that your manuscript has been deemed suitable for publication in PLOS ONE. Congratulations! Your manuscript is now with our production department. 

With kind regards,

on behalf of

Dr. Oscar Millet 

Academic Editor

PLOS ONE